# Trends in collisions and traffic mortality rates in Mexico City: A comparison of six data sources

Martha Laura Herrera Ortiz[1], Carolina Pérez Ferrer[1]*, Carolina Quintero Valverde[1], Luis Chías Becerril[2], Armando Martínez Santiago[2], Héctor Daniel Reséndiz López[2], D. Alex Quistberg[3], Tonatiuh Barrientos Gutiérrez[1]

1 Center for Research in Population Health, National Institute of Public Health, Cuernavaca, Morelos, México, 2 Geography Institute, National Autonomous University of Mexico, Mexico, 3 Urban Health Collaborative, Drexel University, Philadelphia, Pennsylvania, United States of America

* carolina.perez@insp.mx

## Abstract

### Introduction

Improving data quality is an international recommendation to advance road safety. Mexico City has several sources of road safety data which are used interchangeably, no study has analyzed their strengths and limitations.

### Objective

We aimed to compare the trends of four indicators (total collision rate, collision resulting in injury rate, fatality rate and mortality rate) across six open access datasets of Mexico City, between 2015–2022, and to discuss their differences, strengths and limitations.

### Materials and methods

Datasets consulted were from: police records, emergency calls, an insurance company, the justice department, the Institute for Forensic Science and vital registrations for the period 2015–2022. We descriptively compared rates and their trends and estimated percentage changes from the start to the end of the period.

### Results

The collision, collision resulting in injury and mortality rates varied greatly across datasets. Trends over time were consistent in direction; they showed a decline in collisions and deaths from 2015 to 2020 and an increase from 2020 to 2022. However, the magnitude of the change was very different across datasets.

**Data availability statement:** All relevant data are within the manuscript and its Supporting Information files.

**Funding:** This work is part of the Salud Urbana en América Latina (SALURBAL)/ Urban Health in Latin America project funded by the Wellcome Trust (https://wellcome.org/); award number 205177/Z/16/Z. DAQ was partially supported by the Fogarty International Center of the National Institutes of Health (https://www.fic.nih.gov/) under award number K01TW011782. The funders had no role in study design, data collection and analysis, decision to publish, or preparation of the manuscript. The content is solely the responsibility of the authors and does not necessarily represent the official views of the National Institutes of Health nor the Wellcome Trust.

**Competing interests:** The authors have declared that no competing interests exist.

## Conclusions

None of the datasets was comprehensive enough to provide a full picture of road safety in Mexico City. Differences between datasets may be related to the methodology used to report and register collisions and to the reach and remit of each institution. Our results highlight the need for a more comprehensive data information system for road safety in Mexico City and across the country. We call on researchers, practitioners and policy makers to use available data sources responsibly and to be transparent about their limitations until we progress to a unique source of information.

## Introduction

Having access to good quality, reliable road traffic deaths, injuries and collisions data is crucial to document the nature and magnitude of the problem, monitor trends and evaluate the effectiveness of interventions [1]. It is also important to help build political will to prioritize road safety. For this reason, improving data quality is an international recommendation for road safety [1,2]. This includes, standardizing definitions (e.g., what is meant by fatal victim), and ensuring that road safety information systems are comprehensive, simple, easy to use and consistent with national and international norms [3]. Yet frequently cities rely on imperfect, single-source data to monitor road traffic collisions and deaths [1,4,5]. Data are collected by different sectors as required for their day-to-day functioning, for example law enforcement, transport and health [1]. Single-source data is often subject to bias according to the remit and reach of the sector and/or agency that collects them, hampering decision-making at the local level.

Mexico City is a megacity with a population of 9.3 million and a vehicle fleet of 6.4 million. Road safety is of great concern, since the mortality rate from road traffic collisions was 11 per 100 000 in 2021, which is five times higher than the rate observed in cities such as Rio de Janeiro or Bogotá [6,7]. The city has open access datasets which capture road safety information including: 911 emergency calls [8]; police reports [9]; the justice department [10] and; vital statistics [11]. An additional open access dataset on road traffic collisions is provided by AXA, a private insurance company [12]. These data sources vary in comprehensiveness, methods, and definitions, yet, they have been used indistinctively to assess road safety and to evaluate the impact of interventions in the city [13–16]. This issue has been raised by civil society organizations and by governmental institutions but is missing from the academic literature [5,17].

To date, no study has assessed the differences and similarities across data sources in Mexico City, and no systematic effort has been made to discuss their strengths and limitations. We aimed to describe and compare the trends of four indicators (total collision rate, collision resulting in injury rate, fatality rate and mortality rate) using six open access datasets of Mexico City from 2015 to 2022 and to discuss their strengths and limitations. These six datasets are the most widely used to evaluate road safety in the city both by academics, civil society and government [13–16].

## Materials and methods

### Road traffic collisions and mortality data sources

A road traffic collision is defined as an avoidable event produced by vehicular traffic, in which at least one vehicle intervenes, causing material damage, injuries and/or deaths [18]. We used six data sources to assess road traffic collisions and road traffic mortality (Table 1):

1. Police data processed by the National Institute of Geography and Statistics (ATUS for its acronym in Spanish): ATUS is a national system that collects information on road traffic collisions from police reports. Data follows a process of validation and integration. It has the longest coverage from 1997 to 2023 and it has information of time, location, and characteristics of the incident and people involved [9]. National level data source.

2. AXA insurance company: AXA is one of the largest private car insurers in Mexico City. Claims adjusters, using electronic devices, collect the information at the site of the collision. Data are available from 2015 until 2023 for public use; and the variables available for each collision include date, location, severity of the collision [12]. National level data source.

3. Center of Command, Control, Computer, Communication and Citizen Contact (C5): Data are collected by a 911 call center, emergency alarms and/or video-cameras. Each event is validated on-site by a police officer after a report has been filed. Information about the type of incident, date, and location are available for public use; data are available for the period of 2014–2023 [8]. Duplicate records were dropped in two stages. First, we dropped the records with duplication in all variables, except the serial/identification number variable. Second, records were considered duplicates if they shared the same incident code, geographic coordinates, and the reports were one hour or less apart [14]. City level data source

4. Justice department (FGJ, acronym in Spanish): Data are collected by public prosecutors after a crime, including road traffic collisions, categorizing them as causing damages, injuries, or deaths. This dataset, therefore, includes more severe collisions that are prosecuted as crimes. Information includes date of the collision, location and type of crime which includes homicide and injuries due to negligence associated with traffic. Available for Mexico City only for years ranging from 2015 to 2023 [10]. City level data source.

5. Institute of Forensic Sciences (INCIFO, for its acronym in Spanish): INCIFO is part of the Justice Department of Mexico City. It conducts autopsies in all deaths by accidents, violence, or suspicious causes [19]. Road traffic deaths

**Table 1. General characteristics of data sources.**

| Data source | Source | Available years | Geographic level |
|---|---|---|---|
| ATUS | Public Security and Transit agencies, Public Prosecutor agencies or civil court | 1997–2022 | Municipality |
| AXA insurance company | Insurance adjuster | 2015–2023 | Street |
| Center of Command, Control, Computer, Communication and Citizen Contact of Mexico City (C5§). | Citizen report or C5 agent (police officer) | 2014–2023 | Street |
| General Prosecutor Office of Justice (FGJ before PGJ§) | Public Prosecutor officer | 2015–2023 | Street |
| Institute of Expert Services and Forensic Sciences (INCIFO§) | Public Prosecutor officer or C5 agent (police officer) | 2000–2021 | Street |
| National Institute of Geography and Informatic (INEGI§) | Epidemiological and Statistic of Deaths Subsystem (SEED§). Public and private health centers | 1998–2021 | Municipality |

§ initials in Spanish

include those where a vehicle was involved according to the report from the public prosecutor's office. Data are not publicly available but were obtained through a freedom of information request for the period 2015–2022. City level data source.

6. Vital statistics compiled by INEGI: These data are validated and reported by the National Institute for Geography and Statistics (INEGI, acronym in Spanish) [11]. Information comes from death certificates filed by the Health Sector, which classify deaths using the codes related to road traffic collisions of the International Classification of Disease, 10th Revision (ICD-10) [17]. We selected the following codes: V02 – V04.9, V09.2 – V09.9, V12.3 – V14.9, V19.4 – V19.9, V20.3 – V28.9, V29.4 – V29.9, V30.4 – V39.9, V40.4 – V49.9, V50.4 – V69.9, V70.4 – V79.9, V80.3, V80.5, V81.1, V82.1, V83.0 – V88.0, V89.2, V89.9, Y85.0, Y85.9 [18,19]. We include deaths that occurred in Mexico City. National level data source.

The six data sources consist of a nominal list of collisions or deaths with variable information for each event, as described above. Individual records for collisions, collisions resulting in injury and deaths were aggregated (added) per year and dataset. All data sources had available information for the study period except for AXA that was missing information for October 2021. Only in this case, we imputed the monthly value of collisions, collisions resulting in injury and deaths using data from September 2021.

## Other data sources

To obtain denominators to calculate rates we used three data sources:

Number of registered vehicles in Mexico City: This information was obtained on a yearly basis from INEGI [20].

Number of insured vehicles: The number of insured vehicles in Mexico City per year was obtained from the Mexican Association of Insurance Companies (AMIS) [21].

Total population: Official mid-year population estimates of the city were obtained from Mexico's National Population Council (CONAPO) [22].

## Statistical analysis

Data sources were consulted between 2020 and 2024. The variables: total collisions, collisions resulting in injury, and deaths by road traffic collisions were operationalized as described in Table 1 of the supplementary material. These indicators are the most widely used by public and private institutions to report locally and nationally all information related to road safety in the country. The *collision rate* was defined as the total number of collisions that occurred in a year among the vehicle fleet of that year (for AXA the number of insured vehicles), per 1 000 vehicles. To estimate the *collision resulting in injury rate* we used the number of collisions with at least one injured person among the vehicle fleet of that year per 1 000 vehicles. The *mortality rate* was calculated by dividing the total number of deaths due to traffic collisions among the mid-year population per 100 000 inhabitants. The *fatality rate* was constructed dividing the number of fatal collisions among the total number of collisions (within the same database) that occurred during the same year for every 100 traffic collisions.

We descriptively compared the rates and their trends across the datasets and calculated percent changes considering the start and the end date of data. For data management we used Stata version 14.0., and Microsoft Excel 2021 for tables and figures.

Further, we discuss strengths and limitations of the data and completeness based on: representativeness, mortality definition (whether it captures only deaths at the scene of the collision or beyond) and rank comparing the absolute number of collisions, collisions resulting in injury and deaths among datasets. We also highlight other characteristics of the data which may be strengths or limitations.

## Ethics statement

This study was exempt from revision by the National Institute of Public Health ethics committee on April 3rd, 2017, because it does not involve human subjects. The analysis is based on aggregated secondary data. Data were routinely collected by various institutions, are public and unidentifiable.

## Results

### Number of collisions and deaths

Table 2 presents the absolute number of collisions, collisions resulting in injury and deaths across datasets. Large differences can be observed across datasets over the period 2015–2022 (data in S2 Appendix Table 2). Focusing specifically on 2022 we observed the following:

**Total collisions (2022)**: C5 reported 87,365 collisions, compared with 7,595 in ATUS. C5 recorded 11 times more collisions than ATUS, while AXA recorded 8 times more than ATUS.

**Collisions resulting in injury (2022)**: C5 again reported the most (36,117), while ATUS reported the fewest (1,648). The proportion of collisions resulting in injury ranged widely from 12% (AXA) to 45% (FGJ).

**Table 2. Total collisions, collisions resulting in injury and mortality in Mexico City from 2015 to 2022.**

| Source | 2015 | 2016 | 2017 | 2018 | 2019 | 2020 | 2021 | 2022 |
|---|---|---|---|---|---|---|---|---|
| **Total collisions** | | | | | | | | |
| ATUS | 12,337 | 11,449 | 12,321 | 11,656 | 10,673 | 6,549 | 6,743 | 7,596 |
| AXA§ | 91,281 | 90,062 | 77,023 | 63,679 | 57,552 | 28,534 | 41,548 | 58,879 |
| C5 | 109,658 | 77,493 | 75,959 | 75,896 | 75,533 | 56,841 | 71,939 | 87,362 |
| FGJ | 9,001 | 8,622 | 8,230 | 8,071 | 8,314 | 6,280 | 7,143 | 8,747 |
| **Collisions resulting in injury** | | | | | | | | |
| ATUS | 2,147 | 2,191 | 1,859 | 2,070 | 2,197 | 1,254 | 1,446 | 1,648 |
| AXA§ | 8,840 | 7,951 | 6,752 | 5,292 | 4,368 | 2,104 | 3,972 | 7,489 |
| C5 | 39,320 | 26,271 | 25,615 | 26,534 | 26,321 | 21,290 | 27,574 | 36,117 |
| FGJ | 4,122 | 4,067 | 3,701 | 3,491 | 3,644 | 2,759 | 3,180 | 3,924 |
| **Fatal collisions** | | | | | | | | |
| ATUS | 200 | 220 | 193 | 223 | 220 | 142 | 167 | 245 |
| AXA§ | 35 | 30 | 11 | 20 | 17 | 11 | 9 | 10 |
| C5 | 457 | 351 | 468 | 396 | 431 | 405 | 525 | 713 |
| FGJ | 714 | 629 | 620 | 553 | 598 | 623 | 607 | 732 |
| **Number of deaths** | | | | | | | | |
| ATUS | 210 | 227 | 203 | 240 | 226 | 150 | 171 | 255 |
| AXA§ | 35 | 35 | 12 | 26 | 18 | 18 | 16 | 10 |
| C5 | NA | NA | NA | NA | NA | NA | NA | NA |
| FGJ¥ | NA | NA | NA | NA | 603 | 631 | 615 | 728 |
| INCIFO | 1021 | 887 | 929 | 914 | 966 | 907 | 970 | 1053 |
| INEGI | 764 | 659 | 640 | 502 | 368 | 587 | 620 | 603 |
| **Vehicle fleet** | 4,997,606 | 5,220,651 | 5,471,904 | 5,801,469 | 6,084,903 | 6,149,969 | 6,235,773 | 6,368,520 |
| **Total insured vehicles** | 2,759,018 | 2,991,345 | 2,572,322 | 2,608,883 | 3,072,268 | 2,813,479 | 2,931,836 | 2,838,907 |
| **Total mid-year population** | 9,058,734 | 9,053,990 | 9,049,086 | 9,041,395 | 9,031,213 | 9,018,645 | 9,003,827 | 8,986,774 |

¥ There were changes in reporting in 2019 therefore previous data are not comparable. § Missing data for October 2021 was imputed using September 2021 data. NA: data not available.

**Fatal collisions (2022):** C5 reported the most (713), while AXA recorded the fewest (10). The proportion of fatal collisions was also very variable ranging from 0.01% (AXA) to 8.5% (FGJ).

**Deaths:** Datasets capturing deaths at the collision site (AXA, ATUS) substantially underestimated fatalities compared with vital statistics (INEGI). ATUS captured 25–61% of the deaths reported by INEGI, whereas AXA captured only 1.7–5.2%. When comparing INEGI vs the Institute of Forensic Science (INCIFO), INEGI appeared to underestimate deaths. INCIFO reported between 30% and 60% more deaths per year than INEGI.

### Trends over time (2015–2022)

**Total collisions.** All datasets showed a decline in total collision rates between 2015 and 2020, followed by an increase through 2022 (Fig 1). AXA reported the largest reduction during 2015–2020 (−69%), while FGJ reported the smallest (−43%). Between 2020 and 2022, the increase in rates ranged from +11% (ATUS) to +51% (AXA).

**Collisions resulting in injury.** Collisions resulting in injury followed the same pattern, with a decline up to 2020 and an increase thereafter (Fig 2). The decline from 2015 to 2020 ranged from −46% in FGJ to −76% in AXA. From 2020 to 2022, the increase varied from +21% in ATUS to +71% in AXA.

**Fatality rate due to collisions.** The fatality rate (Fig 3) increased across all datasets except AXA. Using C5 data, the fatality rate doubled, rising from 0.4 to 0.8 over the study period. In the case of ATUS, the increase was even greater: the rate rose sevenfold, from 0.4 to 3.2.

**Traffic mortality.** Traffic mortality trends (Fig 4) were more variable across datasets, but an increase was observed after 2019–2020 in all datasets except AXA. The magnitude of the increase ranged from +3% in INEGI to +41% in ATUS. AXA was the only source that showed a continuous decline in the mortality rate through 2022. In INEGI, the mortality rate reached its lowest point in 2019 (4.1 deaths per 100,000), then increased in subsequent years but did not return to the initial 2015 level (8.4 deaths per 100,000).

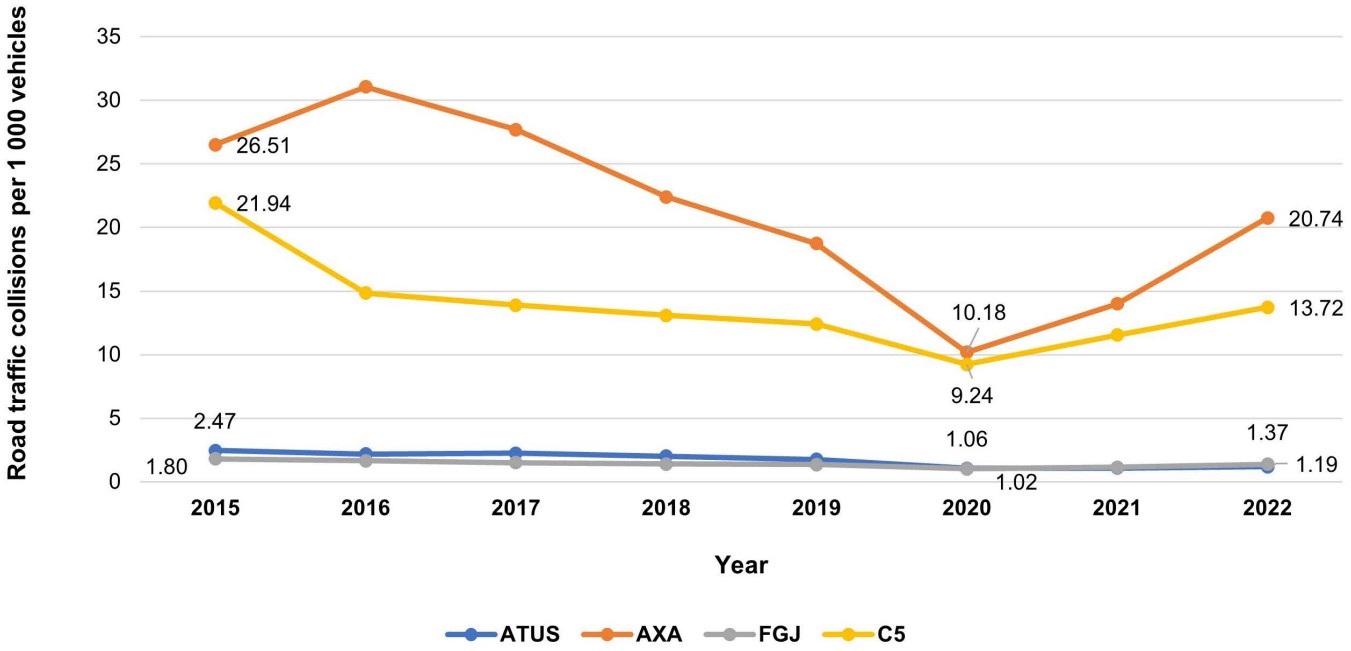

**Fig 1. Total collision rate in Mexico City from 2015 to 2022.** Comparison of four data sources.

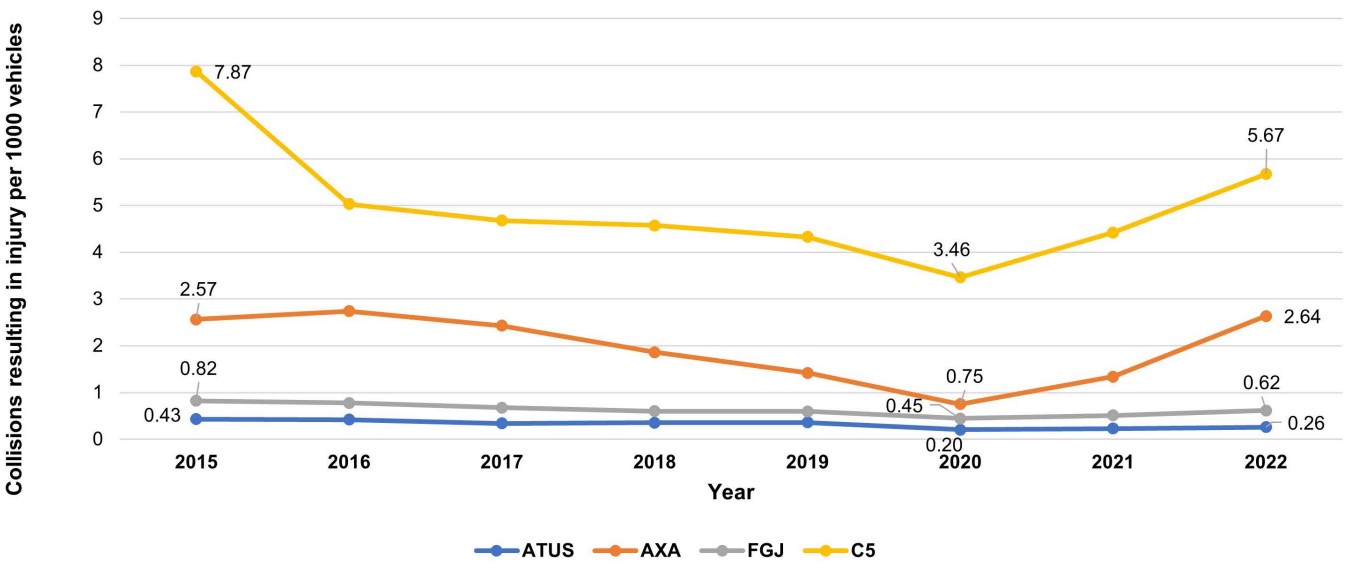

**Fig 2. Collision resulting in injury rate in Mexico City from 2015 to 2022.** Comparison of four data sources.

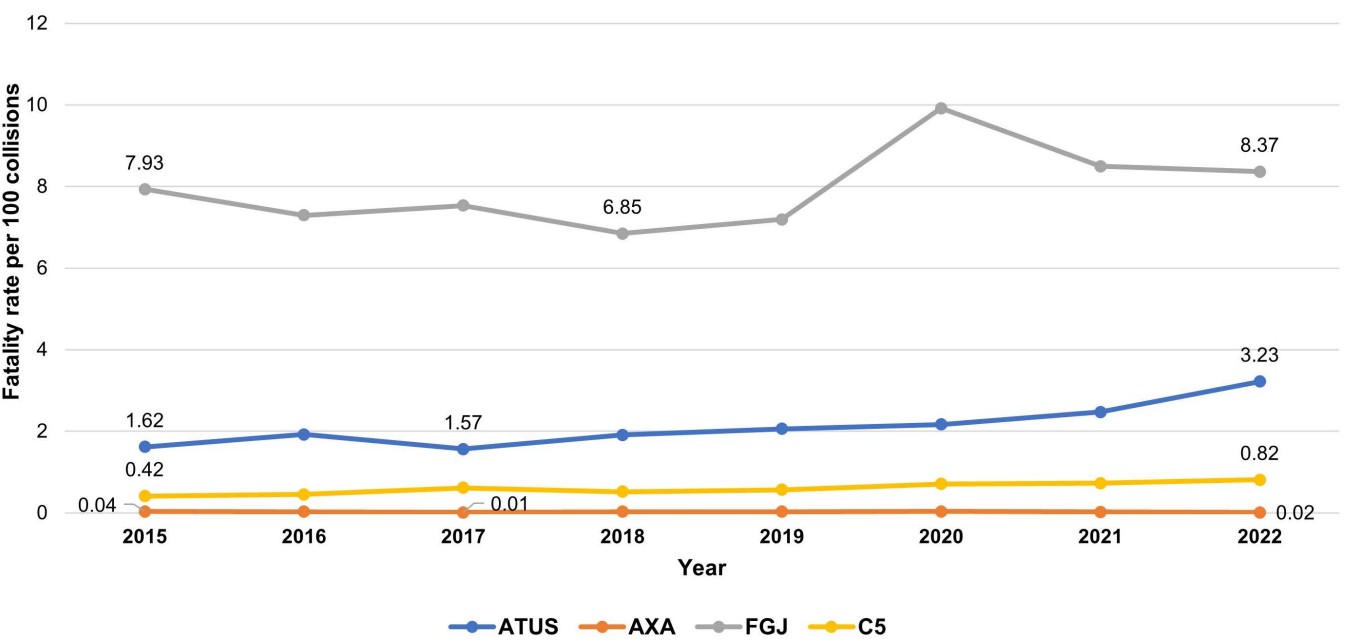

**Fig 3. Fatality rate by road traffic collisions in Mexico City from 2015 to 2022.** Comparison of four data sources.

## Strengths and limitations of the datasets

Table 3 lists some strengths and limitations that were observed in each dataset based on their representativeness, mortality definition and completeness compared to the other data sources. While we do not know the true number of collisions, or deaths, comparing datasets highlights potential underestimation of some indicators. C5 would appear to be the most complete for total collisions and collisions resulting in injury while data from the forensic services captures the largest

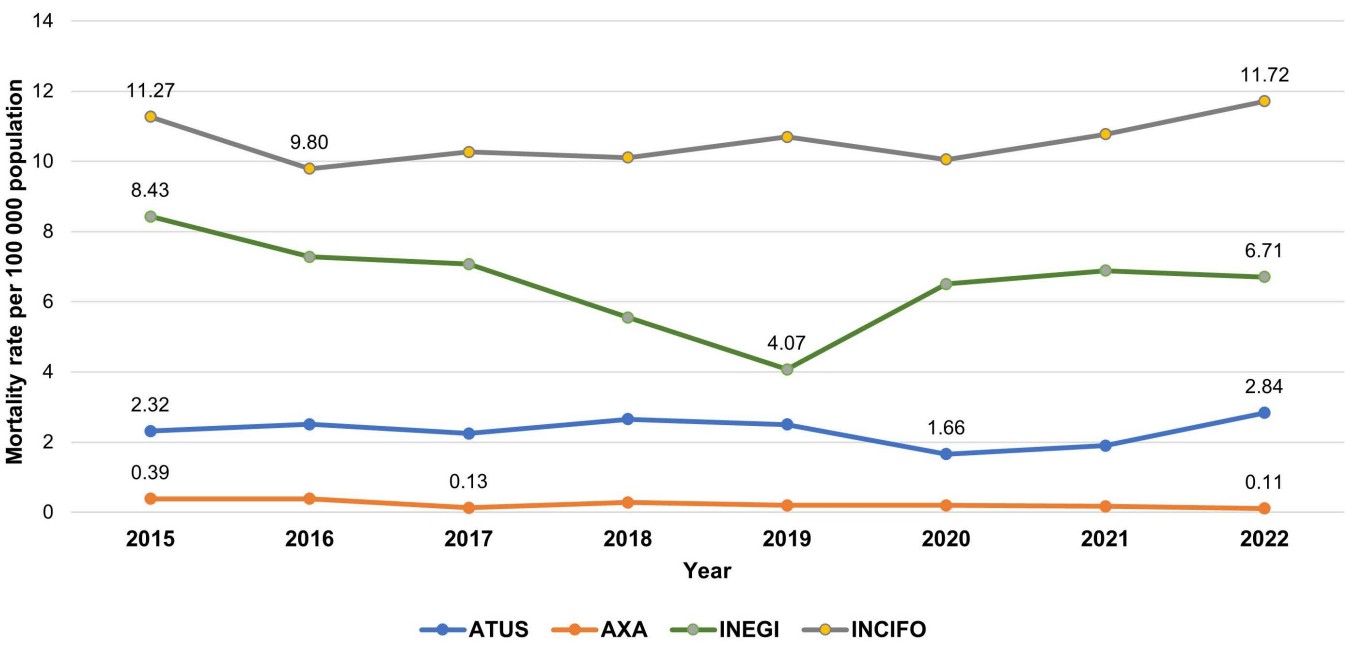

**Fig 4. Traffic mortality rate in Mexico City from 2015 to 2022.** Comparison of four data sources.

number of traffic deaths, followed by the justice department and vital statistics. Datasets such as ATUS, AXA and C5 are not appropriate to estimate mortality or fatal collisions since they only collect information of deaths at the scene. Other strengths and limitations are discussed in the next section.

## Discussion

This study found large differences across datasets in the magnitude of collisions, collisions resulting in injury and mortality rates. Variations may be related to the data collection methods and informants used to report and register collisions and to the reach and remit of each institution [23]. None of the datasets was comprehensive enough to provide a full picture of road safety in Mexico City in terms of collisions, but good information is available on traffic deaths. In terms of the trends of the rates across the 7-year period using different datasets, we found them to be consistent, there was a decline from 2015 to 2019/20 and then an increase in the rates from 2020 to 2022. However, the size of the changes varied considerably across datasets.

Differences observed across datasets for collisions are likely due to different data collection methods and informants. Police reported data underestimate the true number of collisions and are biased towards more severe collisions. This is consistent with evidence that suggests that police registers tend to underestimate collisions due to a lack of capacity for capturing data, loss of follow-up on hospital cases, and incomplete or erroneous data [23–28]. Relying on a larger number of informants, for example citizens, police and emergency services, captures a larger number of collisions as observed with C5. On the other hand, data reported by an insurance company records collisions which are mostly mild and are only representative of insured vehicles (insurance is not compulsory in Mexico).

Differences observed across datasets for mortality are due to the mortality definition used, informant, data collection methodology and misclassification of traffic deaths (poor coding). Datasets that only capture deaths at the scene of the collision (such as AXA, ATUS in Mexico) should be avoided to describe traffic mortality. In this study we found that police recorded about 25% of the deaths classified as traffic deaths in INCIFO, while AXA insurance recorded 1% of the

**Table 3. Strengths, limitations and completeness of data sources.**

| Data source | Strengths | Limitations | Completeness* |
|---|---|---|---|
| ATUS | **Representativeness:** National dataset with longest dataseries (starts 1997). **Other features:** Data are harmonized across states of the country. Publicly available. | **Representativeness:** Includes a subset of all collisions, only those where a public prosecutor or the police arrives to investigate. **Mortality definition:** Deaths recorded at the scene of the collision therefore not useful to estimate mortality rate. | Collisions: ranks 3/4 (highly underestimates) Collisions with injury: 4/4 (highly underestimates) Mortality: 4/5 (highly underestimates) |
| AXA | **Other features:** Data collection is independent of public services or the government in office, and has not changed over time. Collection and completion of the data is linked to insurance claims and payments, which can lead to better care in the collection of information. Collisions are georeferenced. Publicly available. Injuries are classified by level of damage. | **Representativeness:** This is a subset of all collisions because only events occurring among AXA insured vehicles are recorded. Therefore, it is not useful to determine the absolute number of collisions or their rates. Key limitation to study trends is that the correct denominator (AXA insured vehicles) is not available. Collisions are mainly mild, with few injuries and deaths. **Mortality definition:** Deaths are recorded on site, therefore underestimate the true number of deaths among this subset of collisions. | Collisions: ranks 2/4 (highly underestimates) Collisions with injuries: 2/4 (highly underestimates) Mortality: 5/5 (highly underestimates) |
| C5 | **Representativeness:** Larger number of informants (citizens, police, emergency services) which captures a larger number of collisions across Mexico City. | **Mortality definition:** Deaths only recorded at the scene so underestimates real traffic mortality. **Other features:** Causes of collisions have changed over time which presents comparability problems. | Collisions: ranks 1/4 (**better indicator**, may still underestimate mild collisions) Collisions with injuries: 1/4 (**better indicator**) Mortality: N/A |
| FGJ | **Representativeness:** A good indicator for severe collisions that result in deaths in Mexico City. **Other features:** Collisions are georeferenced, and data are publicly available. **Mortality definition:** includes deaths that occur at the scene and beyond. | **Representativeness:** Underestimates the total number of collisions because only collisions prosecuted as crimes are included. **Other features:** Potential misclassification of event date because events may be registered later than they occurred. | Collisions: ranks 4/4 (highly underestimates) Collisions with injuries: 3/4 (highly underestimates) Mortality: 2/5 (**better indicator**) |
| INCIFO | **Representativeness:** The most complete mortality data source for Mexico City. **Mortality definition:** Autopsies determine cause of death for all accidents, violence, and suspicious deaths. | **Representativeness:** Available data for Mexico City, other forensic institutions across the country may have different practices and/or data available. **Other features:** Not publicly available but can be requested through freedom of information requests. | Collisions: N/A Collisions with injuries: N/A Mortality: 1/5 **Best indicator** |
| INEGI – Vital statistics | **Representativeness:** Information at national, state and municipality level. **Mortality definition:** Harmonized data from death certificates filed by the health sector using International Classification of Disease, 10th Revision (ICD-10) CIE-10 codes. | **Other features:** Variable proportion of ill-defined deaths over time because of inconsistencies in coding. This has implications for trends analysis (may show an artificial decline if coding worsens over time). | Collisions: N/A Collisions with injuries: N/A Mortality: 3/5 (moderately underestimates) |

*Rank is based on comparing the total number of collisions, collisions with injury or deaths among datasets with available information; e.g., 2/5 means second highest number of collisions among 5 datasets with available information.

deaths recorded by INCIFO. INCIFO and the justice department data capture more deaths than vital statistics; for example, INCIFO recorded between 30% and 70% more deaths than vital statistics on most years. Discrepancies could have been due to misclassification of deaths in death certificates which INCIFO records as a traffic death some days later after autopsy. We also observe differences in the number of deaths reported by the justice department and the institute of forensic science. A possible explanation for these differences is that the justice department may use unspecific death codes in files which are later confirmed as traffic deaths in the autopsy performed at INCIFO. Based on these findings, the most complete dataset for traffic mortality in Mexico City is INCIFO. However, since this dataset is not public, the next best source is the mortality data from the justice department or vital statistics. For more accurate estimates, vital statistics data can be processed to redistribute ill-defined and partially-defined ICD-10 codes [29].

In Mexico, since 1928, the main source of information on road traffic collisions has been ATUS which, according to our results, vastly underestimates the total number of collisions, injuries and mortality. ATUS data are usually included in international road safety reports by the Pan-American Health Organization (PAHO), and Organization for Economic Co-operation and Development (OECD) [30], thus their accuracy is vital. Subsequently, other sources (AXA, C5, FGJ, SSC, etc.) have become available, and access to information has become easier. This has been positive for various sectors and institutions who have used these data sources to draw conclusions about the status of road safety in Mexico. However, rarely do reports and studies consider the limitations of each dataset. Based on our findings, the availability of various imperfect datasets implies a major responsibility for data management and processing from recording until their ethical and responsible use.

Beyond individuals' responsibility for appropriate data use, our findings also have policy implications. In Mexico, road safety data systems need to be strengthened, aligning with the World Health Organization's (WHO) recommendation that governments and relevant stakeholders prioritize the improvement of road safety data collection systems [31]. This requires developing a multisectoral, secure and adaptable information system capable of generating high-quality data for evidence-based policymaking [1]. Strong data collection systems are essential to measure progress towards the Decade of Action for Road Safety (2021–2030) targets [31].

In Mexico City, the legal framework for a unified and harmonized data source exists [17,32]. For example, in the City's Road Safety Program, activities such as creating an information system with standardized criteria for road safety and a unique identifier for road safety events are included [17]. However, institutional fragmentation has made this very difficult. In 2018, the Institute for Transport and Development Policies (ITDP), published a comprehensive report outlining a series of recommendations for an information system, based on their analysis of the data and interviews with key stakeholders [5]. Many of their recommendations are still relevant seven years later. They include creating a data repository in a dedicated platform where the different institutions that generate information on collisions (ATUS, AXA, C5, FGJ, and others) upload their data, harmonizing data collection across institutions using a standardized, digital instrument and training first responders, police, emergency services at hospitals and other actors in its use. It is also recommended that events have a unique identifier so that it is possible to track the incident from the street to its final outcome. Finally, data assurance is recommended to remove duplicate records and avoid coding errors [5]. Novel technologies, such as the blockchain could be explored to facilitate recording and tracking of events [33].

Intersectoral collaboration and funding are paramount to designing and implementing an information system like this. Relevant sectors include law enforcement, transport, city governments, health, emergency services and insurance companies. The initial implementation phase should include training for users, testing of data collection instruments, procedures and software and gradually rolling out [1]. Quality assurance checks of the data should also be done across all implementation stages. An integrated information system will inevitably be challenging to design and implement. Mexico City could look for successful examples elsewhere and replicate what works [1].

This study has some limitations. To compare data sources, we harmonized variables across datasets. Variables were measured differently and had slightly different definitions. The procedures followed are included in data in S1 Appendix Table 1. Further, data are regularly updated by responsible institutions, so the results can change over time. To address this, we conducted a monthly screening of the sources. An additional limitation is that AXA trends may be the result of fewer insured vehicles (i.e., a smaller denominator). The denominator used may not be sensitive enough to capture changes in vehicles insured by AXA. The strengths of this analysis are that we described and documented the differences in the absolute numbers and trends of four indicators using the main datasets of road traffic collisions in Mexico City; this had not been done in the past and sheds light into the limitations of each source but also into their potential uses.

## Conclusions

Mexico City has a variety of data sources to monitor road safety produced by public and private institutions. This study documented large differences in the magnitude of four road safety indicators using six different datasets. We highlight

strengths and limitations of each dataset that should be considered when using them. Having access to good quality, reliable road traffic deaths and collisions data is crucial to monitor trends and to assess the impact of prevention policies. Our results highlight the need for a more comprehensive data information system for road safety in Mexico City and across the country. We call on researchers, practitioners and policy makers to use available data sources responsibly and to be transparent about their limitations until we progress to a unique source of information.

## Supporting information

**S1 Appendix. Table 1. Road traffic collisions variables included.**
(DOCX)

**S2 Appendix. Table 2. Road traffic collision rates in Mexico City from 2015 to 2022.**
(DOCX)

## Acknowledgments

The authors acknowledge the contribution of all SALURBAL project team members. For more information on SALURBAL and to see a full list of investigators see https://drexel.edu/lac/salurbal/team/. SALURBAL acknowledges the contributions of many different agencies in generating, processing, facilitating access to data or assisting with other aspects of the project. Please visit https://drexel.edu/lac/data-evidence for a complete list of data sources.

The findings of this study and their interpretation are the responsibility of the authors and do not represent the views or interpretations of the institutions or groups that compiled, collected, or provided the data. The use of data from these institutions does not claim or imply that they have participated in, approved, endorsed, or otherwise supported the development of this publication. They are not liable for any errors, omissions or other defect or for any actions taken in reliance thereon.

## Author contributions

**Conceptualization:** Carolina Pérez Ferrer, Carolina Quintero Valverde, Luis Chías Becerril, Héctor Daniel Reséndiz López.

**Data curation:** Martha Laura Herrera Ortiz, Carolina Quintero Valverde, Luis Chías Becerril, Armando Martínez Santiago, Héctor Daniel Reséndiz López.

**Formal analysis:** Martha Laura Herrera Ortiz.

**Funding acquisition:** Tonatiuh Barrientos Gutiérrez.

**Investigation:** Martha Laura Herrera Ortiz, Carolina Quintero Valverde.

**Methodology:** Carolina Pérez Ferrer, D. Alex Quistberg.

**Project administration:** Carolina Pérez Ferrer, Tonatiuh Barrientos Gutiérrez.

**Resources:** Luis Chías Becerril, Armando Martínez Santiago, Héctor Daniel Reséndiz López, D. Alex Quistberg.

**Supervision:** Carolina Pérez Ferrer, Luis Chías Becerril, Tonatiuh Barrientos Gutiérrez.

**Validation:** Héctor Daniel Reséndiz López, D. Alex Quistberg.

**Writing – original draft:** Martha Laura Herrera Ortiz, Carolina Pérez Ferrer, Carolina Quintero Valverde.

**Writing – review & editing:** Martha Laura Herrera Ortiz, Carolina Pérez Ferrer, Luis Chías Becerril, Armando Martínez Santiago, Héctor Daniel Reséndiz López, D. Alex Quistberg, Tonatiuh Barrientos Gutiérrez.

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
