## [Decision Letter · Decision Letter 0]

10 Mar 2025

Dear Dr. Perez Ferrer,

Thank you for submitting your manuscript to PLOS ONE. After careful consideration, we feel that it has merit but does not fully meet PLOS ONE’s publication criteria as it currently stands. Therefore, we invite you to submit a revised version of the manuscript that addresses the points raised during the review process.

**ACADEMIC EDITOR:**

I regret to inform you that we are unable to accept your manuscript for publication in its current form. 

The reviewers have provided valuable feedback on your manuscript. Therefore, we recommend that you thoroughly address all the comments provided by the reviewers and revise your manuscript accordingly.

We understand that revising your manuscript may require substantial effort, but we believe that addressing the reviewers' comments will significantly improve the quality and impact of your research. We encourage you to carefully consider all the feedback provided and to make the necessary revisions accordingly.

Please note that your revision may be subject to further review and that this initial decision does not guarantee acceptance.

We look forward to receiving your revised manuscript.

Kind regards,

Claudio Alberto Dávila-Cervantes, Ph.D.

Academic Editor

PLOS ONE

Journal Requirements:

Reviewers' comments:

Reviewer's Responses to Questions

**Comments to the Author**

1. Is the manuscript technically sound, and do the data support the conclusions?

Reviewer #1: Yes

Reviewer #2: No

Reviewer #3: Yes

2. Has the statistical analysis been performed appropriately and rigorously?

Reviewer #1: N/A

Reviewer #2: No

Reviewer #3: Yes

3. Have the authors made all data underlying the findings in their manuscript fully available?

Reviewer #1: Yes

Reviewer #2: Yes

Reviewer #3: No

4. Is the manuscript presented in an intelligible fashion and written in standard English?

Reviewer #1: Yes

Reviewer #2: Yes

Reviewer #3: Yes

Reviewer #1: This paper offers a unique perspective on traffic safety in Mexico City, addressing a gap in existing research through a comparative analysis of various data sources.

Specific recommendations are as follows:

1. The paper utilizes multiple publicly available datasets for analysis and presents a comprehensive trend analysis of indicators such as collision rates, casualty rates, and fatality rates. The data processing methods align with scientific standards; however, it is recommended that the authors provide a more detailed description of the statistical methods, particularly regarding the handling of missing data and outliers. The current explanation of how missing data are addressed is brief, and a clearer description of how missing values are imputed and how potential duplicate records and outliers are managed would enhance the transparency of the analysis.

2. In the results section, the authors highlight significant differences in collision and mortality rates across the datasets. However, some paragraphs are dense with information, and the sentences are long and complex. Simplifying the expression and breaking down the trends for each dataset into separate, more digestible segments would improve readability and clarity.

3. The discussion of the reasons behind the dataset differences covers several aspects (e.g., data collection methods, limitations, etc.). It would benefit from being divided into smaller paragraphs, with each topic addressed separately. This would enhance the organization and flow of the section, making it more accessible to readers.

4. The conclusions could further elaborate on the limitations of the data sources. For instance, while the authors mention the underestimation of deaths in the AXA and ATUS datasets, a deeper analysis of the specific reasons for this underestimation, along with additional background information, would provide a more comprehensive understanding of these discrepancies.

Reviewer #2: My main concern is the research contribution of the paper is marginal and difficult to be identified. It is unconvincing to me the innovativeness and quality of the paper is sufficiently significant to considered to be published in this journal.

Here are main issues:

1. First of all: How does Mexico City law define "road traffic collisions"? An accident is if at least one moving vehicle is involved – is this the case in Mexico City?

2. How is the term "casualties in an accident" defined in your country? Does it refer to a person who died at the scene of the accident, or does it also include deaths occurring within a certain period after the accident (for example, in some countries, it is 30 days after the accident)? If this is also the case in Mexico City, did you take these deaths into account?

2. AXA data definitely cannot be used to assess and analyze the state of safety, given that this database contains only information about vehicles insured by that insurance company. How many such vehicles are there in relation to the total number of vehicles, and are data collected on all accidents (regardless of the consequences)? Why do you consider these data after all?

3. What is the difference between the third and fourth data sources?

4. How do the data from the third and fifth sources differ? In principle, the source of these data is the same – the Justice Department, so how can they differ from each other?

5. What do the codes related to source 6 represent? And based on what criteria were those particular ones selected?

6. What is the difference between insured and registered vehicles? Can a registered vehicle be uninsured, and conversely, can an insured vehicle be unregistered? And why is the number of insured vehicles even considered?

7. Most important: the paper includes a subtitle "Statistical Analysis," but no actual analysis is presented – which statistical method did you use and for what purpose? The title refers to a "trend," but in the paper, apart from the changes in selected indicators from year to year and their percentage differences, we do not see anything else.

8. The paper lacks an explanation of the motivation for the study. Which databases have been used in previous analyses of traffic safety in Mexico City? Why is there no literature review of previous research – what was used, in what way, and what are the shortcomings of used methods (databases)? Are there studies that research similar questions with usage of different databases?

Based on the above issues, I have to reject this manuscript.

Reviewer #3: The manuscript "Trends in collisions and traffic mortality rates in Mexico City, using different data sources" examines the trends in road traffic collisions and mortality rates in Mexico City from 2015 to 2022, utilizing six open-access datasets. The study is significant in improving road safety data quality. However, there are areas that need improvement in the methodology, discussion of results, and conclusions.

1. Although the paper mentions using multiple data sources to compare the trends in traffic accidents and mortality rates, it does not provide a detailed background on why these four specific indicators (total collision rate, injury-related collision rate, fatal collision rate, and mortality rate) were chosen, or why these six datasets were selected. It would be helpful if the authors could clarify the motivation and significance of these choices in the introduction.

2. The paper includes several figures illustrating the trend variations across different datasets. However, the captions are somewhat brief, and the explanations for the data changes are not always clear. For example, the magnitude of changes mentioned in Figures 1 and 2 should be further analyzed to understand the underlying causes of these trends. Additionally, the figure titles and explanations in the main text should be more consistent to avoid confusion for the reader. Furthermore, it is recommended to supplement the analysis with additional statistical methods, such as time-series analysis or regression models, to assess trend changes more comprehensively.

3. In the discussion section, it is recommended to provide specific policy suggestions to help improve the road safety data system in Mexico City, emphasizing the importance of interdepartmental collaboration.

4. In the conclusion, the authors highlight that the current datasets do not provide a comprehensive picture of road safety and emphasize the importance of integrating more comprehensive data. While this viewpoint is valid, the authors should further discuss how to gradually transition to a unified information system and the potential impacts and challenges that such a system may bring.

**Do you want your identity to be public for this peer review?** For information about this choice, including consent withdrawal, please see our Privacy Policy

Reviewer #1: No

Reviewer #2: No

Reviewer #3: No

---

## [Author Response · Author response to Decision Letter 1]

22 May 2025

Please see attached response to reviewers where we have provided a point by point response to all comments. We feel our manuscript has significantly improved and we thank the editor and reviewers for their time.

---

## [Decision Letter · Decision Letter 1]

1 Sep 2025

Dear Dr. Perez Ferrer,

Thank you for submitting your manuscript to PLOS ONE. After careful consideration, we feel that it has merit but does not fully meet PLOS ONE’s publication criteria as it currently stands. Therefore, we invite you to submit a revised version of the manuscript that addresses the points raised during the review process.

We look forward to receiving your revised manuscript.

Kind regards,

Claudio Alberto Dávila-Cervantes, Ph.D.

Academic Editor

PLOS ONE

Journal Requirements:

Additional Editor Comments:

Reviewer #1:

Reviewers' comments:

Reviewer's Responses to Questions

**Comments to the Author**

Reviewer #1: (No Response)

2. Is the manuscript technically sound, and do the data support the conclusions?

Reviewer #1: (No Response)

3. Has the statistical analysis been performed appropriately and rigorously?

Reviewer #1: (No Response)

4. Have the authors made all data underlying the findings in their manuscript fully available?

Reviewer #1: (No Response)

5. Is the manuscript presented in an intelligible fashion and written in standard English?

Reviewer #1: (No Response)

Reviewer #1: Thank you for the authors' careful response to my previous review comments. The authors have made positive improvements in methodology transparency, discussion structure, and analysis of data source limitations, which have enhanced the overall quality of the paper.

However, I still have concerns about the presentation of the results section. While the authors explained the logic of organizing by indicator type rather than by dataset, which indeed helps maintain consistency in comparisons, the issues of high information density and complex sentence structure have not been fundamentally addressed. Key findings are still buried in lengthy paragraphs, requiring readers to extract main information from complex sentence structures. I suggest the authors further simplify the presentation, consider highlighting important numerical findings in a more prominent way, or add subheadings to break up dense information.

Additionally, while the newly added policy recommendations have practical value, the connection between this section and the preceding research findings could be more coherent. I suggest more clearly explaining how these policy recommendations are derived from the research findings to strengthen the logical flow of the argument.

This study fills an important gap in road safety data quality assessment and has good application prospects. I recommend minor revision before acceptance. I hope the authors can focus on optimizing the presentation of the results section to make key findings more easily identifiable and understandable to readers.

**Do you want your identity to be public for this peer review?** For information about this choice, including consent withdrawal, please see our Privacy Policy

Reviewer #1: No

---

## [Author Response · Author response to Decision Letter 2]

22 Sep 2025

Journal Requirements:

19 Sept 2025: Your ethics statement should only appear in the Methods section of your manuscript. If your ethics statement is written in any section besides the Methods, please delete it from any other section

Response: We have added an ethics statement to the methods section.

Response: The reviewer comments did not include a recommendation to cite additional works.

We have reviewed the reference list to ensure that it is complete and correct. No changes to the reference list were made in this revision. To our knowledge we have not cited any papers that have been retracted.

Reviewer #1:

Thank you for the authors' careful response to my previous review comments. The authors have made positive improvements in methodology transparency, discussion structure, and analysis of data source limitations, which have enhanced the overall quality of the paper.

Response: Thank you for your comments.

However, I still have concerns about the presentation of the results section. While the authors explained the logic of organizing by indicator type rather than by dataset, which indeed helps maintain consistency in comparisons, the issues of high information density and complex sentence structure have not been fundamentally addressed. Key findings are still buried in lengthy paragraphs, requiring readers to extract main information from complex sentence structures. I suggest the authors further simplify the presentation, consider highlighting important numerical findings in a more prominent way, or add subheadings to break up dense information.

Response: Thank you for your comment. We have restructured the results section by breaking up paragraphs, adding subheadings and ensuring consistent order in the way we reported the results. Please see results section.

Additionally, while the newly added policy recommendations have practical value, the connection between this section and the preceding research findings could be more coherent. I suggest more clearly explaining how these policy recommendations are derived from the research findings to strengthen the logical flow of the argument.

Response: Thank you. We have edited one of the paragraphs in the discussion to better link the policy implications to the results of our study. It now reads as follows:

…..Based on our findings, the availability of various imperfect datasets implies a major responsibility for data management and processing from recording until their ethical and responsible use.

Beyond individuals’ responsibility for appropriate data use, our findings also have policy implications. In Mexico, road safety data systems need to be strengthened, aligning with the World Health Organization’s (WHO) recommendation that governments and relevant stakeholders prioritize the improvement of road safety data collection systems [31]. This requires developing a multisectoral, secure and adaptable information system capable of generating high-quality data for evidence-based policymaking [1]. Strong data collection systems are essential to measure progress towards the Decade of Action for Road Safety (2021 – 2030) targets .

In Mexico City, the legal framework for a unified and harmonized data source exists [17, 32]. For example, in the City’s Road Safety Program, activities such as creating an information system with standardized criteria for road safety and a unique identifier for road safety events are included [17]……

This study fills an important gap in road safety data quality assessment and has good application prospects. I recommend minor revision before acceptance. I hope the authors can focus on optimizing the presentation of the results section to make key findings more easily identifiable and understandable to readers.

---

## [Editor Report · Decision Letter 2]

23 Sep 2025

Trends in collisions and traffic mortality rates in Mexico City: A comparison of six data sources

PONE-D-25-05805R2

Dear Dr. Perez Ferrer,

We’re pleased to inform you that your manuscript has been judged scientifically suitable for publication and will be formally accepted for publication once it meets all outstanding technical requirements.

Kind regards,

Claudio Alberto Dávila-Cervantes, Ph.D.

Academic Editor

PLOS ONE
---

## [Editor Report · Acceptance letter]

PONE-D-25-05805R2

PLOS ONE

Dear Dr. Pérez Ferrer,

I'm pleased to inform you that your manuscript has been deemed suitable for publication in PLOS ONE. Congratulations! Your manuscript is now being handed over to our production team.

Kind regards,

on behalf of

Mr. Claudio Alberto Dávila-Cervantes

Academic Editor

PLOS ONE